# Untargeted Fecal Metabolomic Analyses across an Industrialization Gradient Reveal Shared Metabolites and Impact of Industrialization on Fecal Microbiome-Metabolome Interactions

Jacob J. Haffner,[a,b] Mitchelle Katemauswa,[b,c] Thérèse S. Kagone,[d,e] Ekram Hossain,[b,c] David Jacobson,[a,b] Karina Flores,[b,f] Adwaita R. Parab,[b,g] Alexandra J. Obregon-Tito,[a,b] Raul Y. Tito,[a,b] Luis Marin Reyes,[h] Luzmila Troncoso-Corzo,[i] Emilio Guija-Poma,[j] Nicolas Meda,[d] Hélène Carabin,[k,l,m,n] Tanvi P. Honap,[a,b] Krithivasan Sankaranarayanan,[b,g] Cecil M. Lewis, Jr.,[a,b] Laura-Isobel McCall[b,c,g]

[a]Department of Anthropology, University of Oklahoma, Norman, Oklahoma, USA
[b]Laboratories of Molecular Anthropology and Microbiome Research (LMAMR), University of Oklahoma, Norman, Oklahoma, USA
[c]Department of Chemistry and Biochemistry, University of Oklahoma, Norman, Oklahoma, USA
[d]Burkina Faso Ministry of Health, Ouagadougou, Kadiogo, Burkina Faso
[e]Centre MURAZ Research Institute, Bobo-Dioulasso, Burkina Faso
[f]Department of Biology, University of Oklahoma, Norman, Oklahoma, USA
[g]Department of Microbiology and Plant Biology, University of Oklahoma, Norman, Oklahoma, USA
[h]Instituto Nacional de Salud, Lima, Peru
[i]Facultad de Medicina, Universidad Nacional Mayor de San Marcos, Lima, Peru
[j]Centro de Investigación de Bioquímica y Nutrición, Facultad de Medicina Humana, Universidad de San Martín de Porres, Lima, Perú
[k]Department of Biostatistics and Epidemiology, College of Public Health, University of Oklahoma Health Sciences Center, Oklahoma City, Oklahoma, USA
[l]Département de Pathologie et Microbiologie, Faculté de Médecine Vétérinaire, Université de Montréal, Saint-Hyacinthe, Quebec, Canada
[m]Département de Médecine Sociale et Préventive, École de Santé Publique de l'Université de Montréal, Montréal, Quebec, Canada
[n]Centre de Recherche en Santé Publique (CReSP) de l'Université de Montréal et du CIUSS du Centre Sud de Montréal, Montréal, Quebec, Canada

**ABSTRACT** The metabolome is a central determinant of human phenotypes and includes the plethora of small molecules produced by host and microbiome or taken up from exogenous sources. However, studies of the metabolome have so far focused predominantly on urban, industrialized populations. Through an untargeted metabolomic analysis of 90 fecal samples from human individuals from Africa and the Americas—the birthplace and the last continental expansion of our species, respectively—we characterized a shared human fecal metabolome. The majority of detected metabolite features were ubiquitous across populations, despite any geographic, dietary, or behavioral differences. Such shared metabolite features included hyocholic acid and cholesterol. However, any characterization of the shared human fecal metabolome is insufficient without exploring the influence of industrialization. Here, we show chemical differences along an industrialization gradient, where the degree of industrialization correlates with metabolomic changes. We identified differential metabolite features such as amino acid-conjugated bile acids and urobilin as major metabolic correlates of these behavioral shifts. Additionally, coanalyses with over 5,000 publicly available human fecal samples and cooccurrence probability analyses with the gut microbiome highlight connections between the human fecal metabolome and gut microbiome. Our results indicate that industrialization significantly influences the human fecal metabolome, but diverse human lifestyles and behavior still maintain a shared human fecal metabolome. This study represents the first characterization of the shared human fecal metabolome through untargeted analyses of populations along an industrialization gradient.

Address correspondence to Cecil M. Lewis, cmlewis@ou.edu, or Laura-Isobel McCall, lmccall@ou.edu.

The authors declare no conflict of interest.

**IMPORTANCE**   As the world becomes increasingly industrialized, understanding the biological consequences of these lifestyle shifts and what it means for past, present, and future human health is critical. Indeed, industrialization is associated with rises in allergic and autoimmune health conditions and reduced microbial diversity. Exploring these health effects on a chemical level requires consideration of human lifestyle diversity, but understanding the significance of any differences also requires knowledge of what molecular components are shared between human groups. Our study reveals the key chemistry of the human gut as defined by varied industrialization-based differences and ubiquitous shared features. Ultimately, these novel findings extend our knowledge of human molecular biology, especially as it is influenced by lifestyle and behavior, and provide steps toward understanding how human biology has changed over our species' history.

**KEYWORDS**  human microbiome, industrialization, mass spectrometry, metabolomics

Metabolites fit as the final stage of biology's central dogma: DNA transcribed into RNA translated into proteins which enzymatically interact, form, and shed into small molecules as part of the biochemical pathways of metabolism (1–3). For this study, we define a metabolite as any small molecule (<1,500 Da) involved in biochemical pathways and the metabolome as the collection of these small molecules within a biological system (3–5). Using the definition from the Human Metabolome Database, these endogenous metabolites (synthesized by the host) are supplemented by exogenous small molecules acquired from external sources, such as cosmetics, medication, dietary sources, and pollution (6). The human metabolome thus contains both endogenous and exogenous metabolites, representing the nexus of genetic and environmental influences (5, 7–9).

Characterizing the fecal metabolome requires an understanding of how it is influenced by different factors, such as industrialization (10, 11). Broadly, industrialization is a series of economic and technological changes relating to the processing and distribution of resources that ultimately cause a shift from agrarian to industrial societies (12, 13). Such changes generally involve an increase in manufactured products compared to agriculture/hunting and other raw products, a greater percentage of workers employed in industrial workplaces over agriculture, and changes in the physical landscape such as increased construction of built environments (14, 15). Industrialization is often linked with urbanization, which refers to social and demographic shifts increasing population size and density within a settlement (14). These processes lead to industrialized-urban populations exhibiting denser populations (14), reduced exposures to nature-derived molecules but increased exposure to human-derived molecules (16–20), an indirect relationship with food sources (21, 22), and dietary shifts (22, 23) compared to nonindustrial rural populations. Moreover, industrialization is associated with significant biological changes, such as reduced microbial diversity (20, 24–26), increased allergic diseases (27, 28) and asthma (29), and heightened susceptibility to illnesses such as inflammatory bowel disease (30–32), although further work is required to definitively show industrialization processes as the primary cause of these changes given that such health conditions have complex causes (33, 34). Investigations into industrially caused fecal metabolomic shifts have identified differences based in amino acids, amines, sphingolipids, and hexoses, among others (23, 34, 35). Some studies detailed human fecal metabolomes by comparing rural and urban populations and found differences in levels of acylcarnitines, amino acids, and short-chain fatty acids (35–37). However, such studies employed targeted/semitargeted metabolomic approaches and/or sampled a single human population (23, 25, 35–37). As a result, these studies do not represent ranges of human diversity and behavior, highlighting the need for broader investigations of the human fecal metabolome in terms of geographic range and chemical space.

We performed untargeted liquid chromatography mass spectrometry (LC-MS)-based metabolomics on 90 fecal samples obtained from six human populations from diverse geographic regions (Fig. 1a; Table 1; Table S1 in the supplemental material).

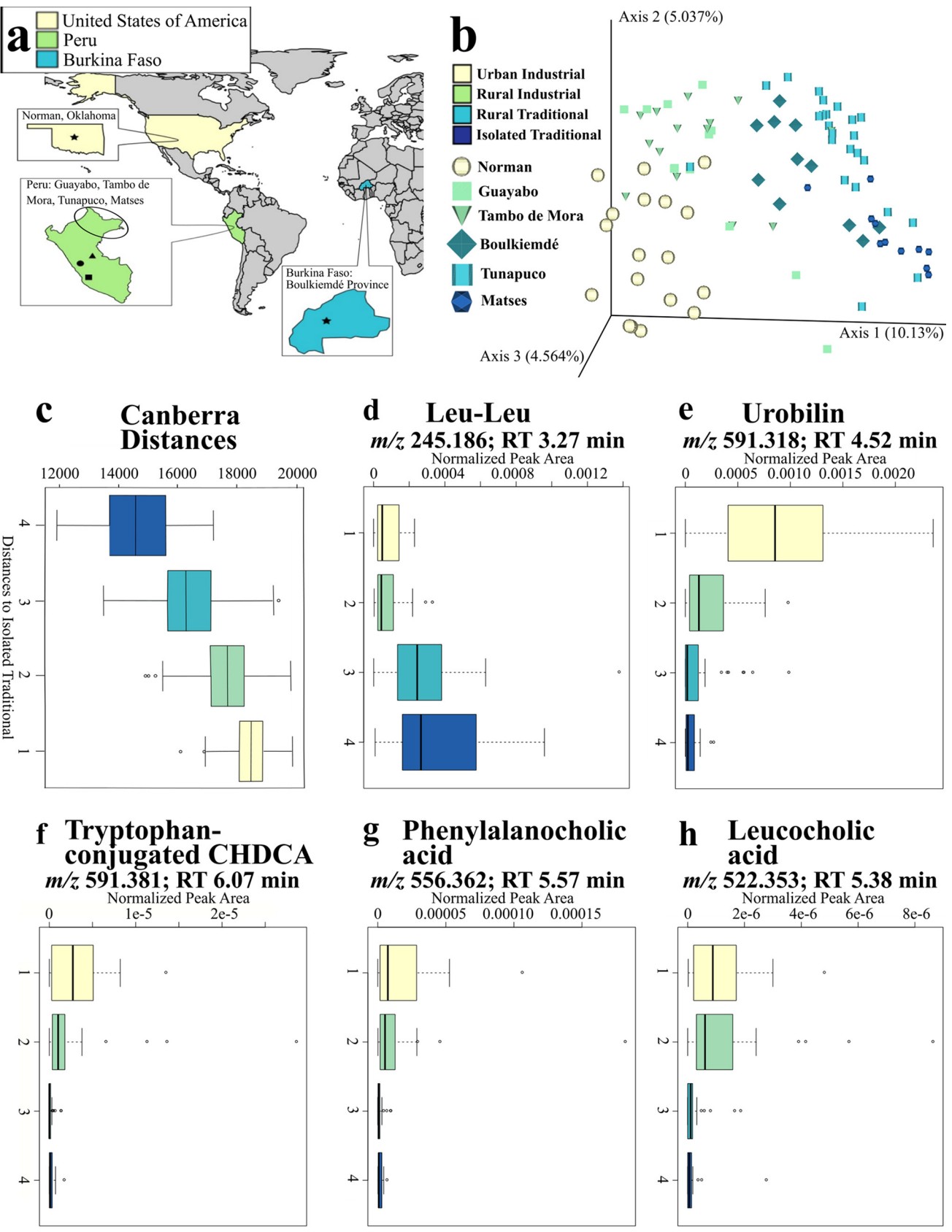

**FIG 1** Fecal metabolomic profiles follow an industrialization gradient. Derived from analyses where sample size (*n*) = 90. (a) Sampling sites. Star on tan background, Norman (*n* = 18); circle on green background, Guayabo (*n* = 12); square on green background: Tambo de Mora (*n* = 14); triangle on

**TABLE 1** Sampled population metadata

| Population | Abbreviation | Geographic origin | Industrialization group | Sample size (n) | Time kept on ice before frozen | Age distribution | | | Sex distribution | |
|---|---|---|---|---|---|---|---|---|---|---|
| | | | | | | 5–17 yrs | 18–44 yrs | 45+ yrs | Female | Male |
| Total | | | | 90 | | 28 | 47 | 15 | 47 | 29 |
| Norman | NO | Norman, OK, USA | Urban industrial | 18 | Within 24 h | 0 | 18 | 0 | 7 | 11 |
| Guayabo | GU | Guayabo, Peru, South America | Rural industrial | 12 | Within 4 days | 4 | 4 | 4 | 8 | 0 |
| Tambo de Mora | TM | Tambo de Mora District, Peru, South America | Rural industrial | 14 | Within 4 days | 5 | 7 | 2 | 7 | 1 |
| Boulkiemdé | BF | Boulkiemdé Province, Burkina Faso, Africa | Rural traditional | 11 | Within 24 h | 0 | 6 | 5 | 6 | 5 |
| Tunapuco | HCO | Andean Highlands, Peru, South America | Rural traditional | 24 | Within 4 days | 13 | 9 | 2 | 13 | 7 |
| Matses | SM | Peruvian Amazon, South America | Isolated traditional | 11 | Within 4 days | 6 | 3 | 2 | 6 | 5 |

These populations included male and female children and adults. Our sampled populations were categorized corresponding with their degree of industrialization, based on lifestyle factors such as dietary practices, built environment, population, etc. (see Materials and Methods for further details on categorization). Importantly, we included two populations with similar degrees of industrialization but from distinct continents, to control for any geographic confounders. This key aspect had not been considered in prior industrialization-focused metabolomics research. Our populations are Norman (USA; urban industrial; 18 samples), Guayabo (Peru; rural industrial; 12 samples), Tambo de Mora (Peru; rural industrial; 14 samples), Boulkiemdé (Burkina Faso; rural traditional; 11 samples), Tunapuco (Peru; rural traditional; 24 samples), and Matses (Peru; isolated traditional; 11 samples).

## RESULTS AND DISCUSSION

Fecal metabolomes of these populations followed an industrialization gradient, where populations exhibited similar metabolomes based on the degree of industrialization determined by principal-coordinate analysis (PCoA; Fig. 1b and c; Fig. S1; permutational multivariate analysis of variance [PERMANOVA] (38) $P = 0.001$, $R^2 = 0.140$; Canberra distance). Moreover, industrialization had a stronger influence on metabolic similarity between populations than geographic origin, age, or sex (Fig. 1c; ANOVA industrialization group $P = 0.046$, effect size [partial Epsilon-squared, eta2] = 0.06; ANOVA geographic origin $P = 0.245$, eta2 = 0.01; ANOVA age $P = 0.4663$, eta2 = 7.29e-3; ANOVA sex $P = 0.5471$, eta2 = 4.99e-3). Delay to initial freezing did impact the overall fecal metabolome (PERMANOVA $P = 0.001$, $R^2 = 0.04$; ANOVA $P = 0.4563$, eta2 = 6.32e-3), but these effects were overshadowed by the influence of industrialization (Fig. 1b). For example, the Boulkiemdé rural traditional and Norman urban industrial samples were frozen within 1 day of collection, but the Boulkiemdé samples clustered strongly with Peruvian rural traditional samples frozen within 4 days of collection (Fig. 1b). Our findings concur with prior studies demonstrating industrialization's role in shaping the human microbiome (39–42), the built environment microbiome (19, 20), the built environment metabolome (19), and the plasma metabolome (25, 43). Additionally, the observation of industrialization outweighing the effects of geographic origin is novel for human fecal

**FIG 1** Legend (Continued)

green background, Tunapuco ($n = 24$); oval on green background, approximate Matses location ($n = 11$); star on a blue background, Boulkiemdé ($n = 11$). The specific Matses location was left unmarked due to privacy concerns. (b) Principal-coordinate analysis (Canberra distance metric) depicts the industrialization gradient, colored by industrialization category and shape-coded by population. Population samples stored in freezer within 1 day of collection (Norman and Boulkiemdé) are increased in size compared to samples stored within 4 days of collection (all Peruvian samples—Guayabo, Tambo de Mora, Tunapuco, and Matses). (c to h) Boxplot axis numbers represent different industrialization groups: 1, urban industrial; 2, rural industrial; 3, rural traditional; 4, isolated traditional. (c) Calculated Canberra distances follow an industrialization gradient, colored by industrialization category. The color key from panel b applies to panels c to h. (d and e) Normalized abundances of representative features identified by random forest analysis differing by industrialization category, (d) Leucyl-leucine (leu-leu), associated with nonindustrialized populations ($m/z$, 245.186; RT, 3.27 min). (e) Urobilin, associated with industrialized populations ($m/z$, 591.318; RT, 4.16 min). (f to h) Normalized abundances of amino acid-conjugated bile acids depict an industrialization gradient. (f) Tryptophan-conjugated chenodeoxycholic acid (CHDCA) ($m/z$, 591.381; RT, 6.07 min). (g) Phenylalanocholic acid ($m/z$, 556.36; RT, 5.57 min). (h) Leucocholic acid ($m/z$, 522.353; RT, 5.38 min).

metabolomics analyses but concurs with findings from human fecal microbiome studies (39–42). To the best of our knowledge, this is the first study to illustrate the industrialization gradient in the human fecal metabolome—the intuitive path for revealing the key chemistry of the distal gut.

To determine the factors driving this clustering of metabolite profiles by industrialization degree, we employed a random forest machine learning algorithm applied to the top 1,000 most abundant metabolite features in our data set (44) (Table 2; Fig. S2a to ad; Data Set S1). After applying a variable importance cutoff of >1.3 to subset the most differential metabolite features, 377 features remained for annotation. A total of 163 (43.1%) metabolite features had compound-level annotations (Data Set S1) according to the Metabolomics Standards Initiative (45). Random forest annotations included glycyl-phenylalanine (mass-to-charge ratio [$m/z$], 223.108; retention time [RT], 0.38 min; amino acid dipeptide composed of glycine and phenylalanine), piperine ($m/z$, 286.144; RT, 5.45 min; plant metabolite common to pepper plants [46]), and isoleucylproline ($m/z$, 228.155; RT, 0.77 min; amino acid dipeptide detected in human urine [47, 48]). When examining the most differential features, two noteworthy annotations were leucyl-leucine ($m/z$, 245.186; RT, 3.27 min; Kruskal-Wallis $P = 8.73e-09$) and urobilin ($m/z$, 591.318; RT, 4.52 min; Kruskal-Wallis $P = 4.45e-07$). Leucyl-leucine (leu-leu) abundance was most associated with nonindustrial populations, while urobilin abundance was strongly associated with industrialized populations (Fig. 1d and e). Leu-leu is a common leucine dipeptide that has not been mentioned in previous industrialization-focused studies of human fecal metabolomes. However, increased abundance of leucine was noted in fecal metabolomes of urban Nigerian adults compared to rural adults (35), contrasting with the nonindustrial association of leu-leu in our data. The second annotated differential metabolite feature, urobilin, is formed from the metabolic breakdown of hemoglobin (49). While previous industrialization-focused fecal metabolomics studies did not report this metabolite, urobilin has been identified as a common metabolite in human urine and fecal metabolomes (50, 51). Importantly, urobilin abundance is affected by host diet and behavior (52), with increased abundance seen in populations consuming diets rich in animal fat, proteins, and carbohydrates (53), such as those seen in industrialized populations. Given the strong association between industrialization, diet, and the metabolome (21, 22, 54), it is likely that some unannotated differential metabolite features represent dietary differences between our sampled populations. Meat and processed food consumption was most frequent in industrialized populations, suggesting that any potential dietary metabolites, such as urobilin, likely originate from these industrialized food sources. One such potential dietary source could be artificial sweeteners, which can strongly influence fecal metabolomes (55). Additionally, the higher consumption of raw vegetable and fruit products in less industrialized communities such as the Matses would also likely drive metabolomic differences. Other potential industrialization-related sources for differential metabolites could include pharmaceuticals and built environment exposure (16, 19, 20) and gut microbiota modulation of dietary metabolite presence/absence (36, 55).

Recent research has revealed novel amino acid-conjugated bile acids that are produced by gut microbiota (56–58) and enriched in patients with inflammatory bowel disease (57). Given the possible association between inflammatory bowel disease and industrialization processes (30–32), we investigated the distribution of these amino acid-conjugated bile acids across our industrialization gradient. Overall, 10 of the 12 total amino acid-conjugated bile acids annotated in this study demonstrated a striking increase with industrialization, despite not appearing in the list of the top 1,000 most abundant features in our data. Such differential amino acid-conjugated bile acids include phenylalanocholic acid (Kruskal-Wallis $P = 1.9e-6$), leucocholic acid (Kruskal-Wallis $P = 1.69e-7$), leucine-conjugated chenodeoxycholic acid (CHDCA) (Kruskal-Wallis $P = 0.04$), tyrosocholic acid (Kruskal-Wallis $P = 7.71e-3$), tyrosine-conjugated deoxycholic acid (Kruskal-Wallis $P = 1.61e-5$), glutamate-conjugated CHDCA (Kruskal-Wallis $P = 1.69e-7$), tryptophan-conjugated CHDCA (Kruskal-Wallis $P = 4.9e-7$), aspartate-

**TABLE 2** Top 30 most differential metabolite features as determined by random forest classifier

| Feature[a] | m/z | RT (min) | P value (Kruskal-Wallis) | Annotation | Details | Predicted ClassyFire/CANOPUS chemical class with posterior probability (%) | Mass difference to reference | Adduct | Analog? | Cosine score |
|---|---|---|---|---|---|---|---|---|---|---|
| 1 | 145.13 | 0.321 | 1.05E-09 | | Part of same subnetwork as feature 1 | | | | No | |
| 2 | 145.13 | 0.322 | 1.62E-09 | | | | | | No | |
| 3 | 159.15 | 0.359 | 3.65E-09 | | | Primary alcohol (71.332) | | | No | |
| 4 | 235.17 | 0.251 | 2.28E-07 | | | | | | No | |
| 5 | 245.19 | 3.274 | 8.73E-09 | Spectral match to leu-leu | In subnetwork with other leu-leu spectral matches | Amino acid derivative (87.591) | 0 | M + H | No | 0.89 |
| 6 | 276.11 | 0.411 | 8.68E-09 | | Part of subnetwork with matches to N-acetylmuramic acid | Organic phosphoric acid and derivatives (59.786) | | | No | |
| 7 | 276.11 | 0.423 | 1.43E-06 | | Part of the same subnetwork as feature 6; also part of a cluster with matches to glycan lacto-N-biose and N-acetylmuramic acid | Organic phosphoric acid and derivatives (59.786) | | | No | |
| 8 | 286.18 | 1.41 | 4.75E-05 | | | Secondary carboxylic acid amide (54.113) | | | No | |
| 9 | 286.18 | 1.677 | 7.10E-06 | | Part of same sub-network as feature 8 | Secondary carboxylic acid amide (54.113) | | | No | |
| 10 | 305.19 | 3.744 | 2.66E-06 | | | Carbamate esters (70.111) | | | No | |
| 11 | 332.07 | 0.36 | 6.36E-08 | | | Aryl chloride (83.961) | | | No | |
| 12 | 363.21 | 1.018 | 1.76E-08 | | | Monosaccharide (59.675) | | | No | |
| 13 | 363.21 | 0.874 | 1.78E-06 | | Part of same subnetwork as feature 12 | Monosaccharide (59.675) | | | No | |
| 14 | 365.19 | 0.514 | 7.39E-09 | | | Monosaccharide (56.026) | | | No | |
| 15 | 379.3 | 4.804 | 4.91E-12 | | | Lipid and lipid-like molecule (53.344) | | | No | |
| 16 | 379.3 | 4.823 | 1.17E-10 | | Part of same subnetwork as feature 15 | Lipid and lipid-like molecule (53.344) | | | No | |
| 17 | 379.3 | 4.804 | 4.25E-12 | | Part of same subnetwork as features 15 and 16 | Lipid and lipid-like molecule (53.344) | | | No | |
| 18 | 379.3 | 4.811 | 1.56E-10 | | Part of same subnetwork as features 15, 16, and 17 | Lipid and lipid-like molecule (53.344) | | | No | |
| 19 | 398.34 | 4.761 | 1.28E-07 | | | Fatty acid ester (60.662) | | | No | |
| 20 | 398.34 | 4.829 | 9.19E-08 | | | Fatty acid ester (60.662) | | | No | |
| 21 | 398.34 | 4.842 | 1.15E-07 | | | Fatty acid ester (60.662) | | | No | |
| 22 | 398.34 | 4.807 | 1.65E-07 | | | Fatty acid ester (60.662) | | | No | |
| 23 | 400.36 | 4.832 | 8.29E-07 | | | | | | No | |
| 24 | 414.34 | 4.493 | 3.73E-09 | | | | | | No | |
| 25 | 414.34 | 4.428 | 1.18E-09 | | | Fatty acid ester (63.169) | | | No | |

**TABLE 2** (Continued)

| Feature[a] | m/z | RT (min) | P value (Kruskal-Wallis) | Annotation | Details | Predicted ClassyFire/CANOPUS chemical class with posterior probability (%) | Mass difference to reference | Adduct | Analog? | Cosine score |
|---|---|---|---|---|---|---|---|---|---|---|
| 26 | 414.34 | 4.379 | 1.07E-10 | | | Fatty acid ester (63.169) | | | No | |
| 27 | 414.34 | 4.428 | 9.24E-11 | | | Fatty acid ester (63.169) | | | No | |
| 28 | 591.32 | 4.516 | 4.45E-07 | Spectral match to urobilin | Part of subnetwork with matches to bilirubin | Fatty acid ester (77.006) | 0 | M + H | No | 0.79 |
| 29 | 593.33 | 4.979 | 3.03E-09 | | | 6-alkylaminopurine (51.054) | | | No | |
| 30 | 597.37 | 5.313 | 3.27E-06 | | | Depsipeptide (68.585) | | | No | |

[a]These features represent the 30 most differential metabolite features based on mean variable importance scores.

conjugated CHDCA (Kruskal-Wallis $P$ = 1.13e-5), histidine-conjugated CHDCA (Kruskal-Wallis $P$ = 6.41e-3), and histidine-conjugated cholic acid (Kruskal-Wallis $P$ = 0.04) (Fig. 1g and h; Fig. S2ae to ap). Interestingly, high abundances of bile acids such as phenylalanocholic acid and leucocholic acid were noted in mice fed high-fat diets (57), which is characteristic of Western industrialized societies (59). The enrichment of these bile acids in our industrialized populations parallels these diet studies, further suggesting a link between diet and the metabolome across industrialization. However, two amino acid-conjugated bile acids, aspartate-conjugated cholic acid (Kruskal-Wallis $P$ = 0.05) and threonine-conjugated CHDCA (Kruskal-Wallis $P$ = 0.4), were not enriched in industrialized populations and did not display any statistically significant differences based on industrialization category. The functional role of these amino acid-conjugated bile acids in health is currently unknown, though our results further support a link between amino acid-conjugated bile acids and industrialization, and possibly to associated diseases.

Our sampled populations are considerably different from each other with strong dietary, behavioral, and geographic differences and, together, represent distinct realms of human experience and diversity. Thus, metabolite features common to these markedly separate populations likely constitute shared components of a human fecal metabolome found in major human groups, even if metabolite abundances vary. Frequency assessment of metabolite features can, however, be strongly influenced by data processing parameters, particularly gap-filing and data filtration. Gap-filling identifies peaks that are present in only some samples and searches for these same peaks at lower intensities in the remaining samples (60). Analyzing non-gap-filled data can artificially increase divergence between groups, while gap-filling may increase similarities between groups (61, 62). Gap-filling is a recommended approach for feature-based molecular networking (61). However, to ensure the greatest transparency, we present the analysis of both gap-filled and non-gap-filled data here.

Analysis of non-gap-filled data identified 8,017 metabolite features with at least one occurrence in each population (27,707 common metabolite features in gap-filled data). Further filtering by occurrences in each population highlighted 7,483 metabolite features in non-gap-filled data found in at least six samples in all populations (23,477 metabolite features in gap-filled data), 2,240 metabolite features in non-gap-filled data found in half of all samples in each population (5,924 metabolite features in gap-filled data), and 1,080 metabolite features in both non-gap-filled and gap-filled data found in every sample across all populations (Fig. 2 for gap-filled data; Fig. S1 for non-gap-filled data). The impact of industrialization on overall fecal metabolome profiles was comparable between gap-filled and non-gap-filled data (compare Fig. 1b to Fig. S1c).

We discuss here the most stringent results based on the non-gap-filled data, while acknowledging that this approach likely misses many features that are actually common across populations, due to analytical considerations. We further filtered out researcher-derived molecules such as N,N-diethyl-meta-toluamide (DEET) from our list of the shared fecal metabolome. These retained common metabolite features included chemical groups such as indoles, steroids, lactones, and fatty acyls (Table S2; Fig. S3). Dipeptides included threonylphenylalanine ($m/z$, 267.134; RT, 0.48 min), valylvaline ($m/z$, 217.155; RT, 0.45 min), and isoleucylproline ($m/z$, 229.155; RT, 0.55 min). Shared bile acids include hyocholic acid ($m/z$, 158.154; RT, 4.78 min; primary bile acid involved with absorbing and transporting dietary fats and drugs to the liver [63]), and lithocholic acid ($m/z$, 323.273; RT, 6.84 min; secondary bile acid commonly found in feces [64]). Fatty acid examples include 3-hydroxydodecanoic acid ($m/z$, 199.169; RT, 7.10 min; medium-chain fatty acid associated with fatty acid metabolic disorders, potentially acquired from the microbial genera *Pseudomonas*, *Moraxella*, and *Acinetobacter* [65, 66]), and palmitoleic acid ($m/z$, 237.001; RT, 6.42 min; fatty acid commonly found in human adipose tissue; also acquired in diet from human breast milk [67]). Additional metabolites include cholesterol ($m/z$, 369.352; RT, 10.5 min; essential sterol found in animals [(6)], methionine ($m/z$, 105.058; RT, 0.33 min; amino acid), and leucine enkephalin ($m/z$, 336.192; RT, 3.21 min; peptide

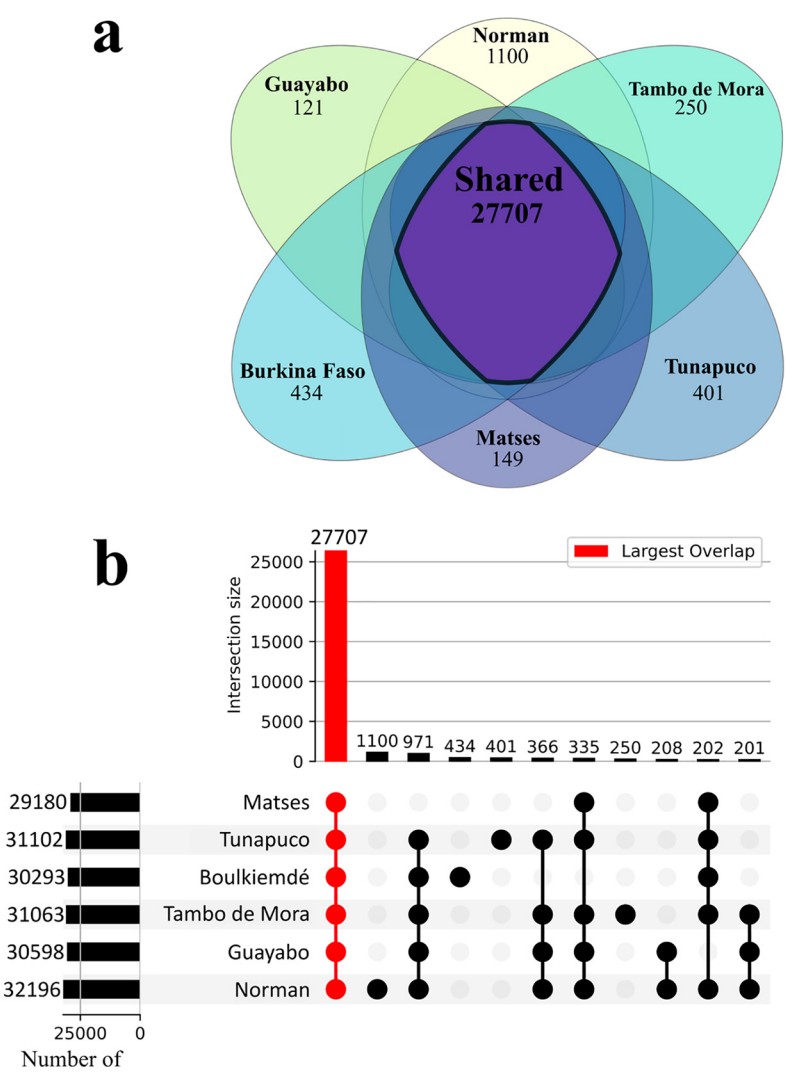

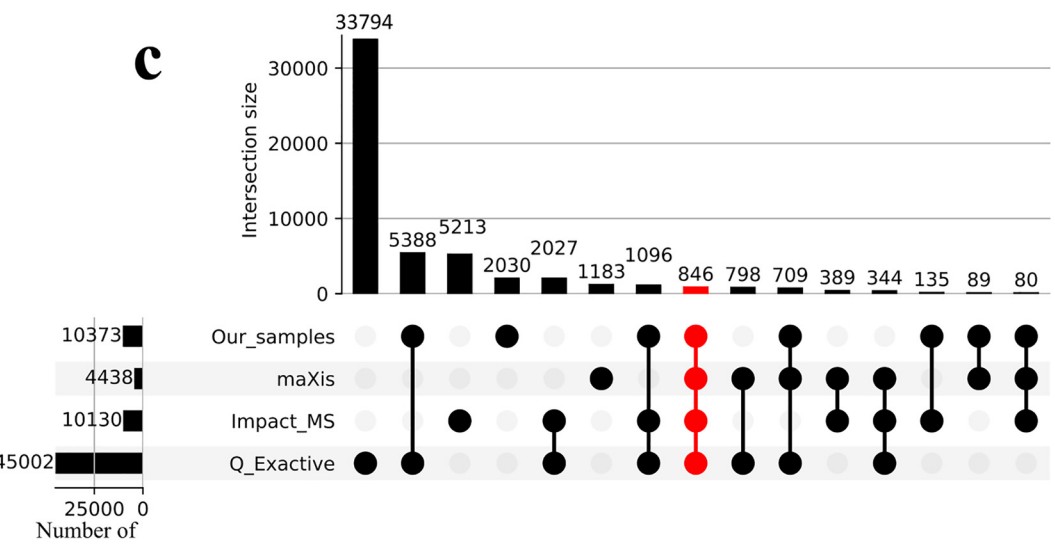

**FIG 2** The shared human metabolome, where $n = 90$ (Norman $n = 18$, Guayabo $n = 12$, Tambo de Mora $n = 14$, Boulkiemdé $n = 11$, Tunapuco $n = 24$, and Matses $n = 11$). (a) Metabolic feature overlap across study populations in gap-filled data. (b) The

naturally produced in animal brains, including humans [6, 68]). While a number of the shared metabolite features listed above provide key biological functions, some metabolites appear to be derived from dietary sources. An example of a metabolite possibly acquired from food products includes conjugated linoleic acid (m/z, 263.24; RT, 6.68 min; commonly found in meat and dairy products [6]).

To explore possible interactions between this shared human fecal metabolome and gut microbiome, we used the neural network platform microbe-metabolite vectors (mmvec) (69). Briefly, mmvec predicts the abundance of metabolites given specific microbial sequences and then estimates conditional probabilities of cooccurrences between the metabolite and microbe being compared. Given the compositional nature of microbiome and metabolomics data (70, 71), mmvec is a robust approach for inferring interactions between gut metabolites and microbes compared to standard correlation analyses (69). Our mmvec analysis used microbial amplicon sequencing variants (ASVs) derived from earlier sample analyses (24) (see Materials and Methods for more details) that were assigned to taxonomic identifications and input to mmvec with our full metabolite feature data set. After subsetting results to our 67 shared annotated metabolites and their major predictive taxa (27 total), several probable interactions between key gut metabolite features and microbes were observed (Fig. 3; Data set S1). For example, five microbial species within the *Sporobacter* genus and one unknown member of the *Anaeroplasmataceae* family were identified as the most influential taxa. Given that five of the six most influential taxa in our data set were *Sporobacter* species, these results suggest a possible connection between these species and the shared human fecal metabolome. Metabolite features such as *N*-acetyl-L-phenylalanine exhibited strong predictive interactions with an unknown *Sporobacter* species (Fig. 3), as shown by high conditional probabilities. Other strong relationships were observed between abrine and another *Sporobacter* species, as well as glycyl-tyrosine and *N*-acetyl-D-mannosamine being strongly driven by the *Anaeroplasmataceae* member. These potential associations had not been noted in previous literature. All in all, these mmvec results suggest clear patterns of predicted interactions between our shared metabolites and gut microbial taxa, but further work is needed to investigate the connections between the shared human fecal metabolome and gut microbiome, especially with regard to the influence of industrialization.

Our novel data estimate a core human fecal metabolome from populations of diverse behaviors and lifestyles. The sample set includes the birthplace of humanity and the last continental expansion of our species, Africa and the Americas, respectively; moreover, the sample set includes hunter-gatherer, subsistence farmer, and industrialized lifeways. A metabolite observed across these geographic regions and among these different lifeways is an estimate of a core metabolome, without implying that it is present in every single individual, similar to definitions used when describing the microbiome (72). However, we do not presume to have captured the complete range of diversity of industrial lifestyles or age groups. To broaden our analysis, we coanalyzed our data with all the publicly available human fecal samples in the Re-Analysis of Data User Interface (ReDU) (51). A total of 5,466 human fecal samples from ReDU were coanalyzed with our 90 samples, resulting in a total of 105,707 metabolite features detected across the coanalysis (Fig. 2c). These data sets contained samples from male and female children and adults. Moreover, the data sets included different MS platforms and different metabolite extraction methods, enabling us to assess the commonality of these metabolites across experimental methods. Within these data sets, 80% of

**FIG 2** Legend (Continued)

UpSet plot of gap-filled data indicates strong similarity of metabolomic profiles. The total number of metabolite features for each sampled population is depicted in rows with the number of overlapping features reported as bar graphs. More features were shared by all population groups than were seen across different group comparisons. The red colored box highlights the intersection of all populations (27,707 total metabolite features). (c) ReDU coanalysis data sets sorted by MS instrument: Thermo Fisher Scientific Q Exactive (n = 696), Bruker Impact (n = 447), Bruker maXis (n = 143). The coanalysis illustrates overlap across the data sets, despite instrumental differences. The colored box highlights the intersection of all data sets (846 total metabolite features).

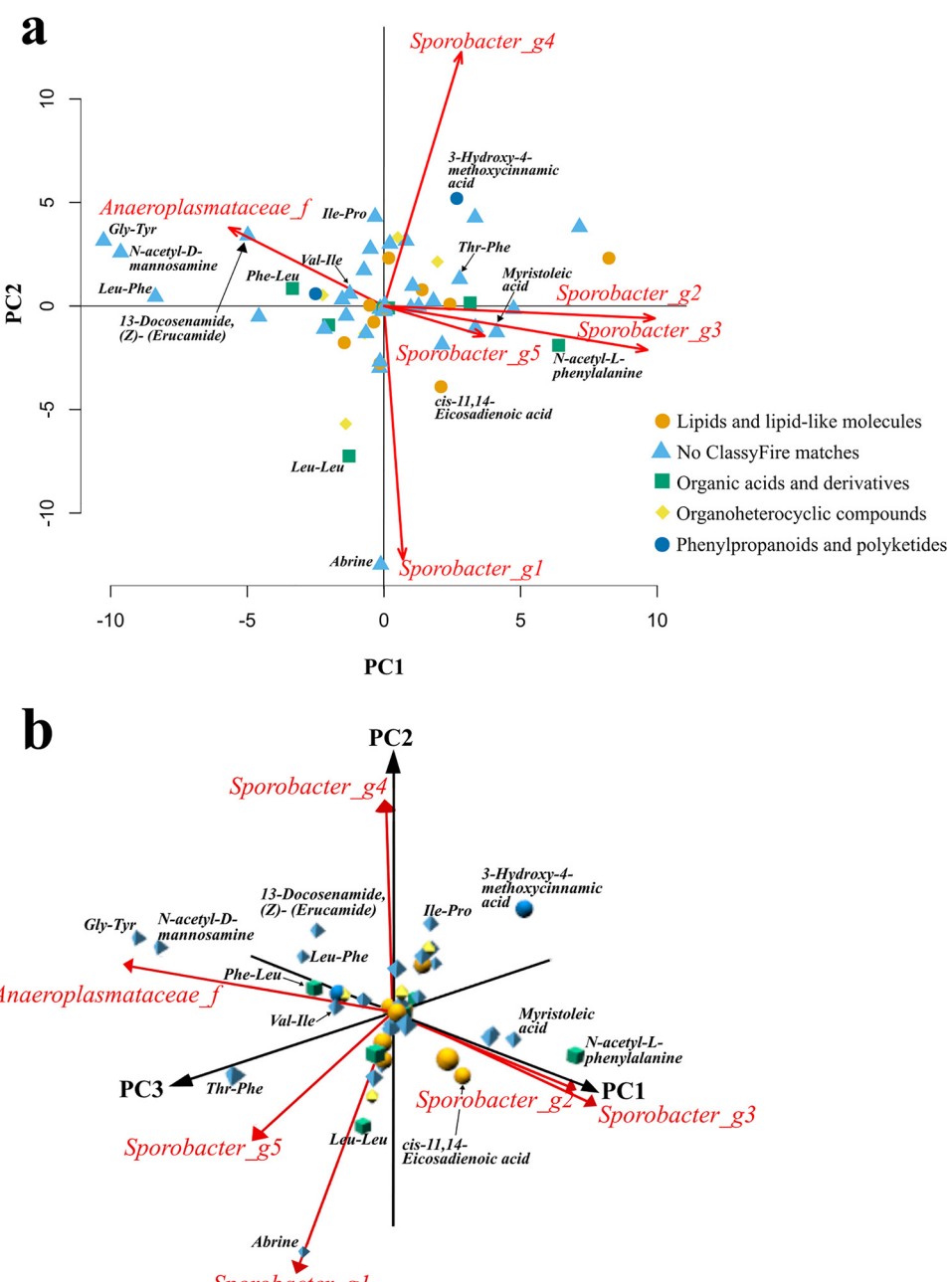

**FIG 3** Principal-component analyses (PCA) illustrate probable metabolite-microbe cooccurrences. Derived from analyses where $n = 90$ (Norman $n = 18$, Guayabo $n = 12$, Tambo de Mora $n = 14$, Boulkiemdé $n = 11$, Tunapuco $n = 24$, and Matses $n = 11$). Metabolite feature placements are based on conditional probabilities produced by mmvec (69). Annotated shared metabolite features from gap-filled data are represented as dots and are shape- and color-coded based on ClassyFire (87) assignments from MolNetEnhancer analyses (88). PCAs include biplots highlighting the most influential taxa across each principal component (PC) represented with red arrows showing their influence along the PCs. Taxonomic assignments were simplified to include unique identifiers for each label, such as "Anaeroplasmataceae_f" representing a read assigned to the family *Anaeroplasmataceae*. Multiple *Sporobacter* genera were identified and were given a "_g" label followed by a number for each instance of *Sporobacter* genera. (a) Two-dimensional representation of shared metabolite-microbe predicted interactions along PCs 1 to 2. Three taxa are represented for each component. Legend from panel a also applies to panel b. (b) Three-dimensional figure of shared metabolite-microbe predicted interactions across PCs 1 to 3. Two taxa are represented for each component.

our shared metabolites were identified in this coanalysis. While these ReDU samples are from data sets containing various instrumental and experimental parameters in addition to wide variation of human diversity and lifestyles, the representation of shared metabolites further highlights the prevalence of a shared human fecal

metabolome across different human populations. Furthermore, we also examined the human fecal metabolome database (HFMDB) (73), which contains 6,810 metabolites identified across multiple data sets, for our annotated shared metabolite features. A total of 65% of our annotated shared metabolite features were present in the HFMDB (Table S2); examples of identified metabolites also found in the HFMDB include palmitoleic acid, hypoxanthine, and xanthosine. However, it should be noted that the HFMDB comprises data derived from various instrumental, analytical, and processing methods (73). The absence of some of our shared metabolites from the HFMDB can be attributed to these methodological differences.

Furthermore, we used the Mass Spectrometry Search Tool (MASST) (58) to search for MS/MS spectra matches to our shared metabolites in public data sets in GNPS (74), Metabolomics Workbench (75), MetaboLights (76), Foodomics (77), and skin trace evidence (Table S3). These searches report sample types matched with our spectra, such as human, mouse, plant, bacterial, or environmental sample types, as well as matched data set names. Across our 67 shared metabolites, MASST reported a total of 4,485 total data set matches, with an average of 67 total data set matchers per metabolite. Indeed, 61% of our shared metabolites reported more matches to human samples than to other sample types (Table S3), and 79% of our shared metabolites also contained bacterial sample matches, suggesting a possible microbiome origin. Additionally, 39% of our shared metabolites were present in human urbanization gradient studies, and 67% were present in studies with cultured bacteria from the gut microbiome. Similar to HFMDB and ReDU, MASST searches contain data collected through various instrumental and experimental methods, so any absent shared metabolites can be attributed to these differences. Nonetheless, the MASST searches demonstrate the prevalence of our shared metabolites across different databases and MS/MS platforms.

While we were able to reveal a shared human fecal metabolome, only 6.1% of our complete data set had putative compound-level annotations (level 2 according to the metabolomics standards initiative [45]). Of these, 15 were validated using standards, enabling level 1 annotation confidence (45) (Fig. S4) and 28.8% of the data set had annotations based only on chemical class (level 3 of the metabolomics standards initiative [45]). This underscores the need for further annotation of human fecal metabolites, especially from human populations traditionally underrepresented in metabolomic databases. Additionally, it is important to note that samples used for this study were collected at different times and subjected to various preservation treatments and lengths. However, our samples clustered based on industrialization category rather than storage conditions or geographic origin, indicating that any confounding influence from preservation was overshadowed by the effect of industrialization. Moreover, industrialization refers to a suite of features that can influence the metabolome, with diet as a strong candidate (54, 55). Other factors such as demography, genetics, and environment also influence the metabolome with diet, highlighting a need to explore the mechanisms of industrialization's effect on the human metabolome. Furthermore, our sampled populations had unequal sex and age distributions, potentially obscuring any effects caused by sex or age on the fecal metabolome. While our results do not indicate statistically significant differences based on age or sex, further research is needed with samples equally representing sex and age distributions. Full data are freely available on GNPS (74) so they can be of use to other researchers and annotations can continue to expand.

Our results demonstrate how industrialization profoundly shapes human fecal metabolic environments regardless of age, sex, or geographic origin. We also highlight strong commonalities in the fecal metabolome across these distinct populations, representing shared features of a human fecal metabolome represented by endogenous and exogenous metabolites. Based on our definition, these shared chemical components are core to major human groups or populations but are not necessarily found in every human individual or LC-MS analysis, given differences in metabolite extraction or instrumental conditions between studies. Further studies focused on untargeted

analyses of a spectrum of industrial and nonindustrial populations, including past and present humans, can help elucidate the shared human fecal metabolome's ubiquity, its relationship with the gut microbiome, and how processes such as industrialization drive human evolution.

## MATERIALS AND METHODS

**Project design.** Fecal samples from six human populations were analyzed, representing ranges of industrialization. Populations were categorized to reflect various degrees of lifestyle behavior and industrialization, based on diet, access to pharmacies and public markets/stores, housing structure, and population density (19). The grouped categories are urban industrialized (highly industrial urban population, typical Western industrialized population; $n = 18$), rural industrialized (industrialized population but primarily rural environment over urban; $n = 26$), rural traditional (rural communities with some industrialization; $n = 35$), and isolated traditional (isolated rural community with little to no industrialization influence; $n = 11$). The study populations include Norman ($n = 18$), OK, USA, a standard Western industrialization population located in the Oklahoma City metropolitan area, Guayabo ($n = 12$), Peru, a large rural town influenced by industrialization, Tambo de Mora district ($n = 14$), Peru, a large rural district influenced by industrialization, Boulkiemdé province ($n = 11$), Burkina Faso, with some industrialization influence, Tunapuco ($n = 24$), a traditional rural community located in the Andean Highlands with minimal industrialization influence, and the Matses ($n = 11$), an isolated traditional hunter-gatherer community from the Peruvian Amazon (Fig. 1; Table 1; Table S1). All populations contained both males and females of ages 3 to 77. Individuals under the age of 3 were excluded from analyses because gut microbiomes do not stabilize and resemble adult microbiomes until after age 3 (42, 78).

**Populations.** Fecal samples from Norman, OK, USA, were analyzed for this project ($n = 18$), representing western industrial lifestyles and diets. Norman residents live in the Oklahoma City metropolitan area, exemplifying a highly industrialized environment. Self-reported diets generally consisted of regular dairy consumption plus processed and/or prepackaged foods such as canned vegetables. Additionally, regular meat consumption was common to Norman individuals compared to our other sampled populations. Due to the strongly industrialized setting and greater consumption of meat, dairy, and processed food products, this population was categorized as urban industrial.

We also selected fecal samples from the Guayabo ($n = 12$) and Tambo de Mora ($n = 14$) populations, which practice similar lifestyles. These populations exhibit rural lifestyles and diets but are still strongly influenced by industrialization. Both communities have regular access to public markets and pharmacies and live in densely packed areas. Their diets are generally reliant on foods obtained from these markets, as well as local produce and livestock. While the Guayabo diet commonly consists of maize with some meat and dairy consumption, the Tambo de Mora population relies more on fish, due to their proximity to the Peruvian coastline. Consumption of processed foods is common for both communities, albeit less so than Norman individuals. Because the Guayabo and Tambo de Mora communities exhibit some characteristics of nonindustrial and industrial lifestyles and live in primarily rural settings, these populations were categorized as rural industrial.

The Boulkiemdé ($n = 11$) and Tunapuco ($n = 24$) communities represent the next degree of industrialization in our sampled populations. Although these populations are from Africa and South America, respectively, they practice similar traditional nonindustrial, rural lifestyles and share some features of industrialized populations, such as access to public markets. The Boulkiemdé samples were collected from the Boulkiemdé province of Burkina Faso. This Burkinabé community practices an agricultural lifestyle, usually growing their own crops, raising livestock, and rarely consuming dairy products. Boulkiemdé meat consumption often ranged from once every 1 to 3 weeks or once every 4 to 6 months. Vegetable consumption was high in self-reported diets of Boulkiemdé individuals. Common vegetables included cabbage, okra, eggplant, beans, carrots, potato, manioc, couscous, rice, corn, etc. Processed foods such as canned vegetables were highly rare. Meanwhile, the Tunapuco population have similar traditional agricultural lifestyles, relying on local produce and livestock. Residing in the Peruvian Andes highlands, the Tunapuco people have diets largely consisting of root and stem tubers, bread, and rice. The Tunapuco people occasionally consume animal proteins and dairy products such as cuy, beef, pork, or sheep. Overall, rice, mote, carrots, cabbages, bread, cuy, oca, and potatoes (fermented, dehydrated, etc.) were the most common self-reported foods for the Tunapuco people. Additionally, Tunapuco residents have access to lowland markets, which offer other dietary sources such as fruit (apples, bananas, pineapples, mangos, etc.), depending on seasonal availability. Similar to the Boulkiemdé community, processed foods are rarely consumed by the Tunapuco people. Since both the Boulkiemdé and Tunapuco communities sampled for this project lived in largely rural yet partly industrial environments with diets focused more on raw food products, these populations were grouped as rural traditional.

Our last sampled population is the Matses ($n = 11$). The Matses people practice traditional hunter-gatherer lifestyles, making them unique for this study. Their diet is based heavily on tubers, plantains, fish, and game meat. Specifically, varieties of manioc, plantains/bananas, and fish are staples of the Matses diet., while bushmeat, reptiles, birds, bread, and other crops are less frequent. Dairy and processed foods are very rarely consumed by the Matses community. Due to their location in the Amazonian regions of Peru and unique lifestyles characterized by self-reliant food production over processed foods, the Matses are almost completely isolated from external sociocultural and economic influences such as industrialization, so they were categorized as isolated traditional.

**Sample collection.** Fecal material was deposited into polypropylene containers and then put on ice. Samples were kept in ice while in the field until arriving at research facilities equipped with freezers. The Norman samples were kept in ice after collection and frozen at the laboratory within 24 h. The Peruvian samples were secured similarly to the Norman samples. After collection, samples were stored on ice for 4 days until arriving at Lima, Peru. Samples were frozen and sent to the laboratory in Norman, Oklahoma.

The Norman, Tunapuco, and Matses samples had previously been aliquoted and underwent 16S rRNA gene sequencing for an earlier study (24), using the MoBio PowerSoil DNA isolation kit protocol (full details can be found in the original article [24]). The raw fecal samples were otherwise kept frozen at −80°C until use for this project.

Boulkiemdé samples were collected similarly to Norman and Peruvian samples. After collection, Boulkiemdé samples were frozen at −20°C within 24 h and kept frozen overnight. Samples were thawed the following evening to extract DNA, refrozen at −20°C, and kept frozen until they were shipped to the laboratory in Norman, OK. Upon arrival, 2 g of fecal material was extracted from each sample for anaerobic culturing. Following this 2-g aliquoting, samples were frozen at −80°C until use for this project.

While field conditions mandated different storage protocols, we confirmed that these effects are overshadowed by the industrialization gradient (see Results).

Full metadata with health conditions, such as primary water sources, pharmaceutical consumption, date of latest hospital visit, etc., were collected for the Boulkiemdé samples. However, the Norman and Peru samples had been collected several years before the Boulkiemdé samples and unfortunately lack similar detailed metadata about health conditions. While this metadata cannot be provided for the Norman and Peru samples, the full deidentified metadata for the Boulkiemdé samples are available in Data Set S1.

**Ethics approval and informed consent.** Ethical protocols for community engagement and sample collection were developed through collaboration with representatives and authorities from each sampled region and in accordance with institutional regulations. All Peruvian samples were obtained through community engagement with local and national authorities and with informed consent with consultation from the Center for Intercultural Health of the Peruvian Institute of Health and Peruvian National Institute of Health ethics committee. This project was reviewed and approved by the research ethics committee of the Instituto Nacional de Salud del Peru (projects PP-059-11 and OEE-036-16).

Human fecal samples were collected with informed consent from resident volunteers in central Burkina Faso under the ethics review committee of Centre MURAZ, a national health research institute in Burkina Faso (institutional review board [IRB] ID no. 31/2016/CE-CM). University of Oklahoma IRB deemed this project consistent with US policy 45 CRF 46.101(b) exempt category 4 (OU IRB 6976).

**LC-MS/MS fecal sample preparation.** The sample preparation protocol used for this project was adapted from a global metabolite extraction protocol with proven success (79, 80). Samples were thawed, and 500 $\mu$L of chilled LC-MS-grade water (Fisher Scientific) was added to 50 mg of fecal material. Next, a TissueLyser homogenized samples at 25 Hz for 3 min. Following homogenization, chilled LC-grade methanol (Fisher Scientific) spiked with 4 $\mu$M sulfachloropyridazine as the internal standard (IS) was added, bringing the total concentration to 50% methanol. The TissueLyser homogenized samples again at 25 Hz for 3 min, followed by overnight incubation at 4°C. The next day, samples were centrifuged at 16,000 × $g$ at 4°C for 10 minutes. Aqueous supernatant was then removed and dried using a SpeedVac vacuum concentrator. Dried extracts were frozen at −80°C until the day of MS analysis. Immediately prior to MS analysis, extracts were resuspended in 150 $\mu$L chilled LC-MS methanol:water (1:1) spiked with 1 $\mu$g/mL sulfadimethoxine as a second IS. After resuspension, samples were diluted to a 1:10 ratio. Diluted samples were sonicated using a Fisher Scientific ultrasonic cleaning bath at maximum power for 10 min. Supernatants were spun briefly to remove any particulates and then loaded into a 96-well plate for MS analysis. One well contained only 150 $\mu$L of the resuspension solution to serve as a blank control.

**LC-MS/MS analysis.** LC was performed on a Thermo Fisher Scientific Vanquish Flex binary LC system with a Kinetex $C_{18}$ core-shell column (50 by 2.1 mm, 1.7 $\mu$M particle size, 100 Å pore size). The LC column was kept at 40°C and the sample compartment was held at 10°C. The LC system was coupled to a Thermo Fisher Scientific Q Exactive Plus hybrid quadrupole-orbitrap mass spectrometer for MS/MS analysis. For the LC mobile phase, solvent A was LC-MS-grade water (Fisher Scientific) with 0.1% formic acid and solvent B was LC-MS-grade acetonitrile (Fisher Scientific) with 0.1% formic acid. The elution gradient started at 5% solvent B for 1 min, increased to 100% solvent B until minute 9, held at 100% solvent B for 2 min, dropped to 5% solvent B over 30 s, and held at 5% solvent B for 1 min as reequilibration. Samples were injected in random order with an injection volume of 5 $\mu$L. After elution, electrospray ionization was conducted with a spray voltage of 3.8 kV, auxiliary gas flow rate of 10, auxiliary gas temperature of 350°C, sheath gas flow rate of 35, and sweep gas flow of 0. Capillary temperature was 320°C, and S-lens radio frequency (RF) was 50 V.

MS1 scan range was 100 to 1,500 $m/z$, MS1 resolution was set to 35,000, and the MS1 automatic gain control (AGC) target was set to 1e6. MS1 data were obtained in positive mode and MS2 data were obtained using data-dependent acquisition. In each cycle, MS/MS scans of each of the five most abundant ions were recorded. Both MS1 and MS2 injection times were set at 100 ms. MS2 resolutions were set to 17,500, the MS2 AGC target was set to 5e5, and the inclusion window was set to 2 $m/z$. MS/MS was conducted at an apex trigger of 2 to 8 s and an exclusion window of 10 s. MS/MS collision energy gradually increased from 20 to 40%.

Authentic standards also underwent LC-MS/MS analysis to validate metabolite annotations. A total of 15 standards were purchased from AA Blocks (hyocholic acid, 13-docosenamide), AvaChem (lenticin), Biosynth (bilirubin, N-acetylmuramic acid, fructosyl-L-lysine), BLD Pharm (N-palmitoylglycine, trans-ferulic acid), ChemScene (leucine enkephalin), LGC Standards (L-saccharopine), Sigma-Aldrich (L-abrine, N-acetyl L-phenylalanine, enoxolone, octadecanamide, lithocholic acid, paraxanthine), and VWR (nicotinamide

*N*-oxide). Each pure standard was diluted to 100 $\mu$M, 50 $\mu$M, 10 $\mu$M, 5 $\mu$M, and 1 $\mu$M concentrations. All standards (and their five dilutions) were analyzed according to the same LC-MS/MS parameters as the original samples. Additionally, fecal extracts with the highest abundance for each standard were reanalyzed as part of the same LC-MS/MS batch to ensure that standard peaks were present in samples and to prevent confounding from retention time shifts caused by the gap between initial data acquisition and annotation validation.

**Data analysis and processing.** MSConvert v3.0.19014 (81) converted raw data files to mzXML format in preparation for data processing via feature-based molecular networking (FBMN) (82). MZmine v2.33 (83) was used to identify MS features for all samples (Table S4). All non-gap-filled analyses were performed using parameters identical to those of the gap-filling steps in MZmine, with the exception of the gap-filling step. After feature filtering, only features with abundance three times greater than the abundance of blanks were retained in these analyses. Total ion current (TIC) normalization was conducted through R programming language v3.5.3 (84) in Jupyter Notebook (85). FBMN and library spectral database searches were completed using the FBMN workflow in Global Natural Products Social Molecular Networking (GNPS) (74). FBMN GNPS parameters for MS/MS analysis were as follows: precursor and fragment ion mass tolerance, 0.02 Da; minimum cosine score for networking and library matches, 0.7; minimum number of matched MS2 fragment ions for networking and library matches, 4; network topK, 50; maximum connected component size, 100; maximum shift between precursors, 500 Da; analog search, enabled; maximum analog mass difference, 100 Da; precursor window filtering, enabled; 50 Da peak window filtering, enabled; normalization per file, row sum normalization. Results were analyzed by visually evaluating mirror plot similarity, cosine score, and match likelihood. Molecular networking results were exported to Cytoscape v3.7.1 (86) to visualize and analyze networks. Predicted ClassyFire (87) classifications for shared metabolites were derived using the MolNetEnhancer (88) workflow in GNPS. In addition, select annotations were confirmed using authentic standards (Fig. S4).

MS filtering was performed in MZmine (83). Three separate filtering workflows were done: 6 minimum peaks in a row (half the number of samples in a single population), 45 minimum peaks in a row (half our total samples), and 90 minimum peaks in a row (all samples). After each filtering step, gap-filling was performed using the previous parameters. For the six-sample filtering, additional processing was done in R (84) to remove any features that were not found in at least six samples from each population. The resulting files were also analyzed in GNPS as described above.

Mass Spectrometry Search Tool (MASST) (58) was used to search for data set matches to the MS2 spectra of our shared metabolites. MASST parameters were as follows: parent mass tolerance, 0.02 Da; minimum matched peaks, 4; ion tolerance, 0.5 Da; score threshold, 0.7; top hits per spectrum, 1; selected databases to search, all (GNPS [74], Metabolomics Workbench (75), MetaboLights (76), Foodomics (77), and skin trace evidence); no analog searches; and no unclustered data search.

For 16S rRNA gene sequencing data, we used AdapterRemoval v2 (89) to filter out sequences of <90 bp in length. QIIME1 (90) was used generate ASVs/zero-operational taxonomic unis (zOTUs) using the EzTaxon database (91) for assigning taxonomic identifiers. EzTaxon was selected over other databases such as Greengenes (92) because EzTaxon is regularly updated, and taxonomic identification was not the purpose for utilizing our 16S data. All samples with fewer than 10,000 reads were removed from analyses. Any ASVs detected in fewer than 10 samples with a maximum abundance of <0.01% were also removed. Generated taxon summaries were limited to genus-level identifications. Only ASVs with >0.5% relative frequency were included in mmvec analyses.

**Mmvec and statistical analyses.** Metabolite and microbe feature tables were input to the Quantitative Insights into Microbial Ecology 2 (QIIME2) (93) microbe-metabolite vectors (mmvec) plugin (69). Conditional probabilities were exported to R (84). Conditionals were subset to our 67 annotated shared metabolites, while major taxa were filtered by exploring high conditional probability values. Filtered results were exported to a new table as .csv. Principal-component analyses were run and visualized using the R (84) package pca3d (94). Further figure modifications were done using Inkscape (https://inkscape.org/) v1.2.

Principal coordinate analysis (PCoA) plots were created using Canberra distance metrics from QIIME2 (93) and visualized using EMPeror (95). PERMANOVA (38) via QIIME2 assessed statistical significance for beta diversity measures. Kruskal-Wallis *P* values were calculated in R (84) through Jupyter Notebook (85). Boxplots (Fig. 1c to h; Fig. S2 and S3) and principal-component analyses (Fig. 3) were also generated using R (84) in Jupyter Notebook (85). For these boxplots, the center line represents the median, the upper and lower box lines reflect upper and lower quartiles, whiskers reflect the interquartile range multiplied by one-and-a-half, and outliers are dots. The R packages ggplot2 (96) and rworldmap (97) were used to create Fig. 1a and c to h. The R package effect size (98) provided *P* values for ANOVA effect size. UpSet plots (99) (Fig. 2b and c; Fig. S1b) were created using the Python 3 (100) packages pandas (101), UpSetPlot (102), and matplotlib (103).

To identify metabolite features unique to specific populations or lifestyles, a random forest machine learning algorithm from the R package RandomForest was used in Jupyter Notebook (44). The number of trees increased gradually from five until reaching a plateau from out-of-bag error at 200 trees. SIRIUS v4.4.26 (104) with ClassyFire (87) classification and CANOPUS (105) compound prediction were used to provide class-level annotations for features identified by random forest analysis.

**Data availability.** LC-MS/MS data were uploaded to MassIVE (106) (accession number MSV000084794). GNPS FBMN jobs are available at https://gnps.ucsd.edu/ProteoSAFe/status.jsp?task=505b8b39810c48eb 9f9b65fee7c6bc7b (v23, original analysis with gap-filling), https://gnps.ucsd.edu/ProteoSAFe/status.jsp?task= b76893f1a07e4cb0be3b603c14cea1b2 (v23, gap-filling, primarily used throughout data analysis), and https://gnps.ucsd.edu/ProteoSAFe/status.jsp?task=af7ec76b02ac482bbd2b7ee3a3ccbdc5 (v23, no gap-filling). FBMN jobs for filtered data are available at: https://gnps.ucsd.edu/ProteoSAFe/status.jsp?task=db 26beb51aff418585e6ad0b92f522b7 (six-sample per population filter, gap-filling), https://gnps.ucsd.edu/

ProteoSAFe/status.jsp?task=4693e01a2af740ceb39bfb19720e798d (six-sample per population filter, no gap-filling), https://gnps.ucsd.edu/ProteoSAFe/status.jsp?task=220d1afd0a564ec1818601d3d928d27a (half-sample filter, gap-filling), https://gnps.ucsd.edu/ProteoSAFe/status.jsp?task=d9686d483e5b496299a02750d6a3ec23 (half-sample filter, no gap-filling), https://gnps.ucsd.edu/ProteoSAFe/status.jsp?task=45150c751a8e42eea51f3ea4936aee95 (all-sample filter, gap-filling), and https://gnps.ucsd.edu/ProteoSAFe/status.jsp?task=45150c751a8e42eea51f3ea4936aee95 (all-sample filter, no gap-filling). ReDU coanalysis is available at https://gnps.ucsd.edu/ProteoSAFe/status.jsp?task=cc2c2d20b20d4bd28c22beb777d2782a (coanalysis with all human fecal samples available in ReDU as of 27 August 2021). This study and the associated raw data are available at the NIH Common Fund's National Metabolomics Data Repository (NMDR) website, Metabolomics Workbench (75) (https://www.metabolomicsworkbench.org; study ID ST002320; DataTrack ID 3495; http://dx.doi.org/10.21228/M8N999). MASST search links are provided in Table S3. Instructions for recreating data analyses in R and Python are available as Jupyter Notebook (85) links at: https://github.com/jhaffner09/core_metabolome_2021. 16S data were uploaded to the Qiita database (study ID 13802; also see study ID 1442 for Norman, Tunapuco, and Matses data).

## SUPPLEMENTAL MATERIAL

Supplemental material is available online only.

**DATA SET S1**, XLSX file, 0.1 MB.
**FIG S1**, PDF file, 1.1 MB.
**FIG S2**, PDF file, 1.1 MB.
**FIG S3**, PDF file, 0.9 MB.
**FIG S4**, PDF file, 0.2 MB.
**TABLE S1**, PDF file, 0.1 MB.
**TABLE S2**, PDF file, 0.2 MB.
**TABLE S3**, PDF file, 0.4 MB.
**TABLE S4**, PDF file, 0.04 MB.

## ACKNOWLEDGMENTS

We thank our collaborators at the Communidad Native Matses Anexo San Mateo, Caserío de Tunapuco, Centre MURAZ Research Institute, and the Ministry of Health in Burkina Faso for their collaboration and for opening their communities to our research. We thank Marielle Hoefnagels and students of the OU BioWriting class for their assistance with editing and reviewing the manuscript.

C.M.L., L.-I.M., and K.S. conceived and designed the study. C.M.L., A.J.O.-T., R.Y.T., L.M.R., E.G.-P., and L.T.-C. led Peruvian sample collection and developed ethical guidelines for community engagement. T.S.K. led fieldwork, metadata curation and sample processing in Burkina Faso and contributed to lab work in the United States. D.J. assisted with fieldwork in Burkina Faso and metadata curation and conducted data analysis. L.-I.M. directed all LC-MS/MS experimentation and data analyses. J.J.H., E.H., and L.-I.M. acquired LC-MS/MS data. J.J.H. and L.-I.M. performed LC-MS/MS data analysis with contributions from M.K., A.R.P., and K.F. J.J.H. wrote the manuscript with contributions from L.-I.M. and C.M.L. All authors reviewed the final manuscript.

We declare no conflicts of interest.

This study was supported by grants from the National Institutes of Health (NIH R01 GM089886) and the National Science Foundation (Doctoral Dissertation Improvement Grant 1925579). Financial support was also provided by the University of Oklahoma Libraries' Open Access Fund.

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
