## [Reviewer comments · mSystems]

Untargeted Fecal Metabolomic Analyses Across an Industrialization Gradient Reveal Shared Metabolites and Impact of Industrialization on Fecal Microbiome-Metabolome Interactions

Jacob Haffner, Mitchelle Katemauswa, Thérèse Kagoné, Ekram Hossain, David Jacobson, Karina Flores, Adwaita Parab, Alexandra Obregon-Tito, Raul Tito, Luis Marin Reyes, Luzmila Troncoso-Corzo, Emilio Guija-Poma, Nicolas Meda, Hélène Carabin, Tanvi Honap, Krithivasan Sankaranarayanan, Cecil Lewis, Jr, and Laura-Isobel McCall

Corresponding Author(s): Laura-Isobel McCall, University of Oklahoma

Review Timeline:

Submission Date:	August 1, 2022
Editorial Decision:	September 28, 2022
Revision Received:	October 28, 2022
Accepted:	November 1, 2022

Editor: Heather Bean

Reviewer(s): The reviewers have opted to remain anonymous.

Transaction Report:

DOI: <https://doi.org/10.1128/msystems.00710-22>

September 28, 2022

Dr. Laura-Isobel McCall
University of Oklahoma
101 Stephenson Pkwy
Norman, OK 73019-5251

Re: mSystems00710-22 (Untargeted Fecal Metabolomic Analyses Across an Industrialization Gradient Reveal Core Metabolites and Impact of Industrialization on Fecal Microbiome-Metabolome Interactions)

Dear Dr. Laura-Isobel McCall:

Thank you for submitting your manuscript to mSystems. We have completed our review and I am pleased to inform you that, in principle, we expect to accept it for publication in mSystems. However, acceptance will not be final until you have adequately addressed the reviewer comments, which are attached below in this email.

Preparing Revision Guidelines

Sincerely,

Heather Bean

Editor, mSystems

Journals Department
Reviewer comments:

Reviewer #2 (Comments for the Author):

In this study, the authors carried out an untargeted metabolomics analysis of 90 fecal samples from humans from Africa and the Americas.

The paper is well written (only minor English edits are needed) and the work is very interesting, although obviously observational.

Please consider the following:

- The authors cannot characterize the core human fecal metabolome based on 90 samples. Please tone down any related statements throughout the text.

Reviewer #3 (Comments for the Author):

Haffner and collaborators describe human fecal metabolomic changes associated with industrialization across a sample cohort collected from four geographically distinct locations and with different degrees of industrialization (urban industrial, rural industrial, rural traditional and isolated traditional). Furthermore authors identify core fecal metabolites common across different geographic locations and industrialization gradients and highlight a set of novel amino acid-conjugated bile acids linked to industrialization and previously associated with inflammatory bowel disease. The study is well written and scientific methods are sound, applied and combined in an innovative manner supporting conclusions. Furthermore the sample cohort is very unique and findings are novel and highly interesting. Authors apply several state-of-the-art methods in the metabolomics and microbiome field skilfully (mmvec, GNPS, MASST, ReDU, MZmine) and combine them in an innovative manner to draw conclusions. Furthermore, authors are very exemplary in making their data and code publicly available, making this valuable dataset accessible to the scientific community. Also, authors are very transparent in their data analysis: feature finding within metabolomics is very sensitive to parameter selection. Authors describe both gap-filled and non-gapfilled results, a level of transparency that most metabolomics studies lack. I believe that previous reviewer's comments contributed to a significant improvement of the manuscript and authors incorporated comments more than satisfactorily.

I only have few minor comments:

It is not clear from Figure 1a where the isolated traditional samples were collected (dark blue, Matses, Peruvian Amazon) Authors state that the initial freezing did impact the overall fecal metabolome, but only report PERMANOVA, can they add a PCoA as well for visualization in the SI?

In line 140 authors refer to File S1, however I was not able to find the file in the submitted documents

Authors extensively describe changes in the fecal metabolome associated with industrialization. Given that industrialization likely also comprises changes in diet and a large majority of metabolites in the fecal metabolome is likely diet derived, could authors comment in the manuscript how they think this will influence their results? Do the authors have any information on diet across the different geographical locations and industrialization gradient, how are they different and are they likely to be more similar across rural areas when compared to urban despite geographical differences?

Reviewer comments:

Reviewer #2 (Comments for the Author):

In this study, the authors carried out an untargeted metabolomics analysis of 90 fecal samples from humans from Africa and the Americas.

The paper is well written (only minor English edits are needed) and the work is very interesting, although obviously observational.

Response: We appreciate the reviewer's appreciation of the manuscript. For those remaining edits, we reviewed the manuscript for grammar/writing issues.

Please consider the following:

- The authors cannot characterize the core human fecal metabolome based on 90 samples.

Please tone down any related statements throughout the text.

Response: As requested, we have rewritten sections of the manuscript with a focus on toning down "the core metabolome" language to instead emphasize a characterization of shared metabolomic components between our sampled populations (new main manuscript: lines 194-196 and 323-327; new full mark-up: lines 199-201 and 369-372; and other changes throughout text).

Reviewer #3 (Comments for the Author):

Haffner and collaborators describe human fecal metabolomic changes associated with industrialization across a sample cohort collected from four geographically distinct locations and with different degrees of industrialization (urban industrial, rural industrial, rural traditional and isolated traditional). Furthermore authors identify core fecal metabolites common across different geographic locations and industrialization gradients and highlight a set of novel amino acid-conjugated bile acids linked to industrialization and previously associated with inflammatory bowel disease. The study is well written and scientific methods are sound, applied and combined in an innovative manner supporting conclusions. Furthermore the sample cohort is very unique and findings are novel and highly interesting. Authors apply several state-of-the-art methods in the metabolomics and microbiome field skilfully (mmvec, GNPS, MASST, ReDU, MZmine) and combine them in an innovative manner to draw conclusions. Furthermore, authors are very exemplary in making their data and code publicly available, making this valuable dataset accessible to the scientific community. Also, authors are very transparent in their data analysis: feature finding within metabolomics is very sensitive to parameter selection. Authors describe both gap-filled and non-gapfilled results, a level of transparency that most metabolomics studies lack. I believe that previous reviewer's comments contributed to a

significant improvement of the manuscript and authors incorporated comments more than satisfactorily.

Response: We appreciate the reviewer's praise for the project and manuscript.

I only have few minor comments: It is not clear from Figure 1a where the isolated traditional samples were collected (dark blue, Matses, Peruvian Amazon)

Response: We had originally left the Matses location unmarked due to privacy concerns. However, we have added an oval to indicate an approximate location for the Matses within Peru while maintaining privacy concerns. Figure 1 legend has also been edited to reflect the figure changes (new main manuscript: lines 1016-1020; new full mark-up: lines 1128-1133).

Authors state that the initial freezing did impact the overall fecal metabolome, but only report PERMANOVA, can they add a PCoA as well for visualization in the SI?

Response: We thank the reviewer for their comments. Figure 1b is a PCoA using color and shape codes to distinguish sample categories, as well as size differences for populations frozen within 1 day (larger) vs. within 4 days (smaller). To improve this clarity, the sizes for the samples frozen within 1 day have been increased again. Text in the main manuscript and figure legend have been also been edited to make this distinction clearer for the figure (new main manuscript: lines 124-127 and 1022-1025; new full markup: lines 126-130 and 1136-1137).

In line 140 authors refer to File S1, however I was not able to find the file in the submitted documents

Response: We apologize for the omission and now provide it with our submission.

Authors extensively describe changes in the fecal metabolome associated with industrialization. Given that industrialization likely also comprises changes in diet and a large majority of metabolites in the fecal metabolome is likely diet derived, could authors comment in the manuscript how they think this will influence their results?

Response: As requested, we have included more detailed discussions about diet, industrialization, and the fecal metabolome (new main manuscript: lines 157-167 and 311-315; new full mark-up: lines 161-171 and lines 357-363).

Do the authors have any information on diet across the different geographical locations and industrialization gradient, how are they different and are they likely to be more similar across rural areas when compared to urban despite geographical differences?

Response: We have expanded discussion of the sampled populations' diets with a focus on key similarities/differences between population diets and provided more information about specific dietary products (new main manuscript: lines 357-360, 367-368, 378-382, 385-391, and 396-399; new full mark-up: lines 403-406, 413-414, 424-428, 432-440, and 443-447). We have also included more detailed diet metadata with the new submission in File S1.

November 1, 2022

Dr. Laura-Isobel McCall
University of Oklahoma
101 Stephenson Pkwy
Norman, OK 73019-5251

Re: mSystems00710-22R1 (Untargeted Fecal Metabolomic Analyses Across an Industrialization Gradient Reveal Shared Metabolites and Impact of Industrialization on Fecal Microbiome-Metabolome Interactions)

Dear Dr. Laura-Isobel McCall:

Your manuscript has been accepted, and I am forwarding it to the ASM Journals Department for publication. There are two very minor edits that I suggest for the final version, which are provided in the attachment. These edits are at the authors' discretion.

For your reference, ASM Journals' address is given below. Before it can be scheduled for publication, your manuscript will be checked by the mSystems production staff to make sure that all elements meet the technical requirements for publication. They will contact you if anything needs to be revised before copyediting and production can begin. Otherwise, you will be notified when your proofs are ready to be viewed.

Publication Fees:

If you would like to submit a potential Featured Image, please email a file and a short legend to mSystems@asmusa.org. Please note that we can only consider images that (i) the authors created or own and (ii) have not been previously published. By submitting, you agree that the image can be used under the same terms as the published article. File requirements: square dimensions (4" x 4"), 300 dpi resolution, RGB colorspace, TIF file format.

We recognize that the video files can become quite large, and so to avoid quality loss ASM suggests sending the video file via <https://www.wetransfer.com/>. When you have a final version of the video and the still ready to share, please send it to mSystems staff at mSystems@asmusa.org.

Sincerely,

Heather Bean
Editor, mSystems

Journals Department
Figure S2: Accept

Figure S3: Accept

Table S2: Accept

Table S3: Accept

Figure S4: Accept

Table S4: Accept

File S1: Accept

Table S1: Accept

Figure S1: Accept